# Associations between Physical Activity Patterns, Screen Time and Cardiovascular Fitness Levels in Swedish Adolescents

**DOI:** 10.3390/children8110998

**Published:** 2021-11-03

**Authors:** Karin Kjellenberg, Örjan Ekblom, Cecilia Stålman, Björg Helgadóttir, Gisela Nyberg

**Affiliations:** 1Department of Physical Activity and Health, The Swedish School of Sport and Health Sciences (GIH), 114 86 Stockholm, Sweden; orjan.ekblom@gih.se (Ö.E.); cecilia.stalman@gih.se (C.S.); bjorg.helgadottir@gih.se (B.H.); gisela.nyberg@gih.se (G.N.); 2Department of Clinical Neuroscience, Karolinska Institutet, 171 77 Stockholm, Sweden; 3Department of Global Public Health, Karolinska Institutet, 171 77 Stockholm, Sweden

**Keywords:** organised sports, moderate-to-vigorous-physical activity, sedentary activity, accelerometers

## Abstract

Cardiovascular fitness (CVF) has been associated with cardiovascular risk factors in adolescents. CVF levels are determined by non-modifiable and modifiable factors; one modifiable factor is physical activity (PA). There is a lack of studies investigating the associations between PA patterns and CVF and how gender, parental education, BMI status and country of birth are associated with CVF. The aim of this study was to explore the cross-sectional associations between PA patterns and CVF in Swedish 13–14-year-old adolescents. CVF was estimated using the Ekblom-Bak submaximal test, data on PA patterns were collected using hip-worn accelerometers and a questionnaire. The mean CVF was 44.8 mL/kg/min in girls (n = 569) and 55.5 mL/kg/min in boys (n = 451) *p* < 0.01. The results showed a significant association between participation in organised sports (β = 3.32 CI: 2.14, 4.51, β = 4.38, CI: 2.80, 5.96), MVPA (β = 0.07, CI: 0.04, 0.11, β = 0.07, CI: 0.03, 0.11), a high proportion of SED (β = −0.47, CI: −0.70, −0.25, β = −0.41, CI: −0.64, −0.18) and CVF in girls and boys, respectively. More than five hours of screen time on weekdays was associated with lower CVF (β = −2.32 CI: −3.92, −0.71 in girls and boys β = −2.82, CI: −5.14, −0.50). While causal relations remain unknown, these findings could be relevant when designing future interventions with the aim to improve CVF.

## 1. Introduction

Studies have suggested that high cardiovascular fitness (CVF) is associated with fewer cardiovascular risk factors in children and adolescents [1,2,3]. Further, longitudinal data indicate that CVF in adolescents is significantly associated with cardiovascular risk factors in adulthood [4]. CVF can therefore be used as an important health indicator in children and adolescents [5]. An age-based cut off of a healthy CVF corresponds to 35 mL/kg/min for girls and 42 mL/kg/min for boys, 13 years of age [6]. More studies are needed to explore the predictors of CVF in different adolescent groups and contexts.

There are many factors that influence CVF, such as age [7], body composition [8], maturity [9] and a genetic component [10], resulting in a large variation in CVF in the pediatric population [1]. The preponderance of data indicates a gender difference in CVF, where girls have been suggested to have lower fitness levels compared to boys, especially when related to total body mass [11]. These differences are accentuated after puberty, making gender and maturation an important factor to consider when estimating cardiovascular fitness in adolescence. One study has shown that the circulatory response to acute exercise in pre-pubertal boys is more similar to the response seen in women than men [12].

Although some variation in CVF is determined by genetic factors, CVF levels can be altered by participation in regular and intense physical activity (PA). The difference in CVF between adolescents involved in regular, intense PA is most evident in post-pubertal children [13,14]. Experimental data indicate that even intense exercise in pre-pubertal children has limited effect on CVF levels, if any [15].

Regular PA during childhood and adolescence is crucial as there are indications that PA habits established early persist during adulthood [16,17]. A longitudinal study in Norway concluded that adolescents who participated in organised sports had more time in leisure-time PA in young adulthood compared to those who did not participate [18]. A Finnish longitudinal study found similar results, and also concluded that those who participated more frequently had higher odds of being active as an adult (odds ratios between 5 and 6) compared to those who did not participate [19]. However, the majority of adolescents are physically inactive (i.e., do not meet the PA recommendations of 60 min of MVPA per day) [20]. Studies investigating the association between different PA behaviors and fitness levels have found associations between PA and fitness levels [21,22,23,24]. For example, there are indications that participation in organised sports are associated with higher performance in over-ground running tests [25].

However, the relative contribution of other aspects of the PA pattern remains largely unknown. Inconclusive evidence exists regarding relations between time spent sedentary (SED) and CVF. Joensuu et al. [23] found no associations between SED and CVF in 9–15-year-olds, whereas Marques et al. showed that a combination of meeting the PA recommendations and spending less time in SED was associated with higher fitness [21]. The authors concluded that time spent in MVPA contributed more to fitness levels than SED [21]. Martinez-Gomez found similar results, but only in girls [26]. There is also inconclusive evidence regarding the association between screen time and CVF [27,28].

The aim of this study was to investigate the cross-sectional association between CVF and components of the PA pattern (time in MVPA, light PA, LIPA, SED), screen time, participation in organised sports, meeting the PA recommendation for Swedish 13–14-year-olds. Further, to establish the CVF levels in different groups of adolescents depending on body mass index (BMI), parental education, country of birth, pubertal maturation, and gender.

## 2. Materials and Methods

This study is part of the larger cross-sectional study Physical Activity for Healthy Brain Functions in School Youth conducted between September–December 2019 with the overall aim to investigate the association between PA and brain health.

### 2.1. Recruitment

All schools (n = 558) with students in grade 7 within a proximity of two to three hours of Stockholm, Sweden were invited to participate in the study. The invitation was sent by email to the principle and vice principals. Small schools (<15 students in grade 7), with a sport profile or with a student population not speaking Swedish were excluded. A total of 84 schools accepted the invitation, five declined, and one was excluded after being invited as the student population could not speak Swedish. To achieve the planned sample size of approximately 1000 students, 40 schools of varying size, geographical, and parental education background were included in the study where six dropped out before the data collection (due to time constraints). As the required sample size of schools was exceeded after the first invitation, nonresponsive schools were not pursued. The recruitment is described in more detail elsewhere [29].

### 2.2. Ethical Statement

All participants had written student and parental consent. The study was ethically approved by the Swedish Ethical Review Authority (dnr: 2019-03579) and conducted in accordance to the Declaration of Helsinki.

### 2.3. Data Collection and Measures

The participating students came with their teachers and classmates to the research facilities at the Swedish School of Sport and Health Sciences. The measures included in this study were anthropometric measurements, a questionnaire, self-reported puberty, a submaximal cycle ergometer test and accelerometry. After the visit, a questionnaire was sent out to the students’ parents by email or mail. The students received a SEK 300 (USD 35) gift card as compensation, whereas school staff and parents did not receive any compensation.

#### 2.3.1. Cardiovascular Fitness

Cardiovascular fitness (CVF) was estimated using the Ekblom-Bak test, a submaximal cycle ergometer test, described in more detail elsewhere [30]. Prepubertal boys in Tanner stage 1–2 were assessed using the equation for females [12]. Change in heart rate response to change in submaximal rate of work was used to estimate maximal oxygen consumption and expressed as relative values (liters of oxygen per minute, L/min) or (milliliters of oxygen per kilogram of body weight per minute, mL/kg/min).

#### 2.3.2. Physical Activity Patterns

Physical activity (PA) patterns (time in sedentary, MVPA and LIPA) were assessed using a hip-worn triaxle accelerometer (model GT3X+, Actigraph, LCC, Pensacola, FL, USA) at a sample rate of 30 Hertz. The participants received the monitor during their visit and were told to wear the accelerometer during all awake time, except for water-based activities, for seven days and which was to be sent back in pre-paid envelopes by the teachers. An individual time filter for expected wear-time was created based on the participant’s reported awake/asleep time (extracted from the questionnaire). The accelerometer data were processed in Actilife (v6.13.3) as uniaxial data, using 5 s epoch time intervals. Non-wear time was defined as 60 min of zero counts and no spike tolerance. A minimum of 500 min of wear time was considered a valid day (after excluding nonwear time), and participants with at least three valid days (including at least one weekend day) were included in the analyses. The data was categorized into intensities using the following cut offs: SED 0–99 counts per minute, LIPA 100–2295 counts per minute, and MVPA > 2295 counts per minute [31]. To minimize observation bias, the first day was removed and the following seven days were included in the analyses [32]. The variables presented in this article include minutes per day spent in MVPA, SED, LIPA and represent an average of the whole week. A variable for reaching the physical activity recommendation of 60 min of MVPA was computed. The variable was based on the average MVPA per day over the whole week and dichotomized into meeting the PA recommendation (≥60 min MVPA) or not meeting the PA recommendation (<60 min MVPA). A ratio variable that accounted for both time spent in SED and MVPA was calculated by dividing average time spent in SED by time spent in SED + MVPA time ∗ 100.

#### 2.3.3. Screen Time

Screen time was self-reported by the participants using the following two questions. “During a normal weekday, approximately how much time do you spend using a screen (not including schoolwork) including a cell phone, TV, computer, iPad? (for example, to play games, watch TV, chat, watch serials, YouTube, Snapchat and Instagram)” and “During a normal weekend day, approximately how much time do you spend using a screen (not including schoolwork) including a cell phone, TV, computer, iPad? (for example, to play games, watch TV, chat, watch serials, YouTube, Snapchat and Instagram)”. The answers included not at all, less than one hour, 1–2 h, 3–4 h, 5–6 h and 7 h or more, and were collapsed into ≤2 h, 3–4 h and ≥5 h.

#### 2.3.4. Participation in Organised Sports

Participation in organised sports was assessed via a questionnaire using the following question, “Are you active in any sports club/organization? (e.g., football, swimming, dancing, scouts, gym)?”. To assess level of participation a follow-up question was used “How many times a week do you participate?”. The answers were arranged on a scale from once a week up to four times per week, and five times or more per week and were collapsed into 1–2 times, 3–4 times and 5 times or more. Those who did not participate in organised sports were used as a reference group for analysis on participation rate.

#### 2.3.5. Anthropometric Measurements

Body weight was measured using a calibrated scale (Tanita BC-418, Tanita corporation, Tokyo, Japan) and height was measured using a stadiometer (SECA 5123, SECA Weighing and Measuring Systems), the results were rounded to the closest 0.1 kg or mm. A continuous BMI variable was calculated using body mass (kg) divided by height (m) squared and BMI was also categorized into four categories defined according to the International Obesity Task Force [33]. BMI standard deviation score (sds) was calculated according to Karlberg et al., 2001 to account for gender and age [34]. Body composition, in this case body fat percentage, was measured with bioelectric impedance analyses (BIA) using a Tanita body composition analysis scale (Tanita BC-418, Tanita corporation, Tokyo, Japan) [35].

#### 2.3.6. Pubertal Maturation

Tanner stage drawings were used as a self-reported measure of pubertal maturation. The participants used two drawings to indicate their pubertal development, one for pubic hair and in boys for genital development and in girls for breast development. This method has been shown to be an acceptable measure to distinguish between pre-puberty and puberty [36] which was the purpose in this study. An average score of the two Tanner drawings were used [37] and also divided into gender based quartiles groups. The mean Tanner score was 2.6 and 2.1 in the first quartile, 3.5 and 3.0 in the second quartile, 4.0 and 3.6 in the third quartile and 4.58 and 4.6 in the fourth quartile, for girls and boys respectively.

#### 2.3.7. Demographics

Gender and country of birth were self-reported by the participants via the questionnaire. Country of birth was dichotomized into Sweden and outside Sweden. Parental educational level was self-reported by the parents in the questionnaire and was dichotomized into ≤12 years and >12 years of education. The highest level of education attained by either of the parents was used.

### 2.4. Statistical Analyses

Data were analysed using STATA/SE version 17.0 [38]. Descriptive statistics are presented using mean, standard deviation for continuous variables and proportions for categorical variables. To compare the difference between boys and girls, as well as different adolescent groups, independent *t*-tests and one-way analyses of variance (ANOVA) with Bonferroni post hoc tests were used for numerical variables and chi-square for categorical variables.

Multiple regression models were used to assess the association between PA patterns and CVF. Considering that the data was collected from clusters (schools), the effect of the school on the outcome variable (CVF) was investigated by including the school as a random term in an empty mixed model. Only 6% of the variance in CVF was explained by the school and therefore regular regression models were used.

As CVF and PA were significantly different between boys and girls, all models were stratified on gender. Further, as multicollinearity was a problem among the PA variables, separate models were used. The assumptions for multiple linear regression were checked, none of the assumptions were violated with the exception of equal variances (Breusch−Pagan test > 0.05). Therefore, robust estimates were used in the multiple regression models to produce robust standard errors. The level of statistical significance was set at *p* < 0.05 and 95% confidence intervals are in bold in the tables to indicate significance.

## 3. Results

Table 1 shows the descriptive characteristics of the sample. A total of 1556 students were invited and 1139 participated in the study (73% response rate). In the final sample 51% were girls, 72% had parents with more than 12 years of education, 86% were born in Sweden and 80% were underweight or had normal weight. One participant indicated gender “other” and was not included in the analyses that were stratified by gender or used gender as a confounder.

There were 1020 students with a valid fitness test, 52% girls and 48% boys, of these 884 had both a valid fitness test and a valid accelerometer reading (i.e., had worn the accelerometer for at least three days, including one weekend day) and were included in the analyses. No significant difference in CVF was seen between the participants who had valid accelerometer readings and those who did not. However, when stratifying by gender, girls who had a valid accelerometer reading also had higher CVF (mean difference of 2.3 mL, *t*(567)= −2.32 *p* = 0.02). The group without a valid fitness test had significantly lower BMI (mean difference of 1.72, *t*(1131)= 4.90 *p* < 0.01) and significantly higher MVPA (mean difference of 7.4, *t*(901) = −3.2 *p* < 0.01).

Approximately 30% of the participants reached the PA recommendations, with more boys than girls meeting the recommendation (mean difference of 12%, *p* < 0.01). On an average day, boys spent significantly more time (5 min) in mean MVPA and in mean LIPA (2 min) compared to girls, and significantly less mean time (15 min) being sedentary. There were no significant difference in sport participation between boys and girls, or between how frequently they participated in the organised activities (see Table 1).

On weekdays, approximately 32% of the sample had up to two hours of screen time, compared to only 16% on weekends. There was a significant difference in screen time between weekdays and weekend days (chi2 = 441.60, *p* < 0.01).

The mean CVF in the whole sample was 49.5 mL/kg/min, where boys had significantly higher CVF than girls (mean difference of 10.68 mL/kg/min (*t*(1018) = −19.58 *p* < 0.01). A total of 89.4% of the girls, and 91.9% of the boys were above the age-based cut off for a healthy CVF. Table 2 shows unadjusted CVF means in different groups of adolescents. As seen, the adolescents with underweight and normal weight had higher CVF (mean difference of 8.95 mL, *t*(1018) = 12.44, *p* < 0.001), compared to those with overweight and obesity.

The adolescent group with high parental education had significantly higher CVF (mean difference 1.86, *t*(871) = −2.52 *p* = 0.01) where the result was only significant in boys when stratifying by gender. Further, adolescents born in Sweden had a significantly higher CVF (mean difference 1.98, *t*(1011) = 2.16 *p* = 0.03) and this finding only remained significant in girls when stratifying by gender. However, these analyses were not adjusted for BMI, therefore a *t*-test was done to compare the BMI between the groups. The results showed that the group with low parental education had significantly higher BMI sds (mean difference 0.39, *t*(871) = 4.48 *p* < 0.01). Further, adolescents born outside Sweden also had significantly higher BMI (mean difference 0.24, *t*(1011) = −2.19 *p* = 0.03) and when stratifying by gender, this difference was only seen in girls.

CVF levels were also significantly different between the pubertal maturation groups where boys in the higher Tanner quartiles had higher CVF and the opposite was seen in girls. However, adolescents in the higher Tanner quartiles also had significantly higher BMI (*p* < 0.01). Further, girls in the highest quartile had 3.3% more body fat compared to those in the lowest group (*p* < 0.01) and this was not seen in boys.

Table 3 shows the associations between CVF and predictors using linear regression models. All models were tested for gender interaction where no significant interaction effects were found. Country of birth and parental education were not significant as confounders and did not change the results and were excluded in the final model. Tanner stage was also added as a confounder but did not change the results and was therefore not included in the final models. MVPA was used as a confounder in the models including screen time and organised sports.

Positive associations between MVPA and CVF were found in girls (β = 0.07, CI: 0.04, 0.11) and boys (β = 0.07, CI: 0.03, 0.11). The lower physical activity intensity LIPA was also associated with higher CVF levels in girls (β = 0.04, CI: 0.02, 0.07) and boys (β = 0.04, CI: 0.01, 0.06). When examining the correlation between the two, LIPA and MVPA was strongly correlated in boys (r = 0.54) and moderately correlated in girls (r = 0.37).

Adolescents reporting participation in organised sports had significantly more time spent in MVPA, mean difference of 13.23 min (*t*(459) = −7.41 *p* < 0.01) in girls and 8.89 min (*t*(339) = −3.78 *p* < 0.01) in boys. Participation in organised sports was also positively associated with CVF (β = 3.32, CI: 2.14, 4.51 in girls and β = 4.38, CI: 2.80, 5.96 in boys) after adjustment for time spent in MVPA. Figure 1 shows the adjusted mean CVF by weekly participation rate in organised sports obtained from model 3. Participating in organised sports at a lower participation rate ≤2 times per week was not significantly associated with higher CVF levels compared to those students who did not participate at all. However, a positive association was seen between the higher participation rate >2 times per week and CVF.

Further, time in SED was negatively associated with CVF in both girls and boys. To account for MVPA, yet avoiding multicollinearity, the proportion of SED divided by MVPA and SED was tested in an identical model. This ratio was significant for girls (β = −0.47, CI: −0.70, −0.25 and boys (β = −0.41, CI: −0.64, −0.18).

As seen in Table 3, five hours or more of screen time per day on weekdays was negatively associated with CVF in both boys and girls. In girls, there was a negative association with >2 h of screen time on weekdays. However, no significant association was seen on weekends. When comparing the groups with low and high screen time on weekdays, boys who reported ≤2 h of screen time also had more MVPA (mean difference 3.6 min *p* < 0.01) and less SED (mean difference of 50.0 min *p* < 0.01), compared to those who reported ≥5 h. In girls, there was no significant difference in MVPA, and SED was only significantly higher in those reporting ≥5 h of screen time (mean difference of 21 min, *p* = 0.03). When using SED as a confounder, the negative association between weekday screen time and CVF remained significant for girls and boys with ≥5 h of screen time (β = −2.26, CI: −3.86, −0.65 in girls and β = −2.68, CI: −5.00, −0.367 in boys).

## 4. Discussion

The aim of this study was to investigate the associations between different PA patterns (MVPA, LIPA, SED), screen time, participation in organised sports, meeting the PA recommendation and CVF in Swedish 13–14-year-olds. Further, to establish the CVF levels in different groups of adolescents depending on BMI, parental education, country of birth, pubertal maturation, and gender.

The main findings were that participation in organised sports was associated with higher CVF and the association was stronger for those who participated more than twice per week. Further, more than five hours of screen time on weekdays was negatively associated with CVF, in girls this was seen also in those with 3–4 h of screen time. MVPA were positively associated with CVF and a high proportion of SED was associated with lower fitness.

The CVF means in our sample were 44.8 mL/kg/min in girls and 55.5 mL/kg/min in boys. The majority (89.4% of the girls, and 91.9% of the boys) were above the age-based cut off for healthy, which corresponds to 35 mL/kg/min for girls and 42 mL/kg/min for boys, 13 years of age [6]. These findings were somewhat similar to Ortega et al. who found that 8.6% of the boys and 19.6% of the girls were below the cut off. Although these findings were based on the old cut off, which corresponds to 38 mL/kg/min for girls and 42 mL/kg/min, which could explain the larger proportion of girls below the cut off [39]. These proportions could be considered surprisingly high since only a small proportion of adolescents meet the PA recommendation (30.2% in our sample).

However, many previous studies in this population have used other tests to assess CVF, such as the the 20m shuttle run which makes any comparisons of estimated CVF difficult. One study in 14–16-year-olds reported girls having a CVF of 40.2 mL/kg/min and boys 51.0 mL/kg/min [39]. Comparing the values obtained from different age groups of adolescents using different test methods should be done with caution, especially considering the impact of factors such as age, gender and maturation status. However, the data do not indicate large differences in CVF values between this and previous investigations.

There has been a lack of studies measuring CVF levels in Swedish adolescents during the last decade. A previous trend study of CVF from 1987 and 2001 indicated a decline in CVF levels. The absolute values from 2001 were 2.2 L/min in girls and 2.5 L/min in boys 16 years of age [11]. This study used the Åstrand test and did not adjust for age and puberty, so any direct comparisons to our results are difficult. However, our absolute values of 2.3 L/min in girls and 2.9 L/min in boys are similar to the previous study, which suggests that there has not been a decline in CVF in Swedish adolescents.

In line with previous studies, we found significant association between BMI and CVF. Further, we found that the differences seen in CVF between parental education and country of birth was due to BMI differences, and when controlling for BMI there were no significant associations to CVF. These findings are similar to a study by Pate et al., where ethnicity was not significantly associated with CVF [22].

Regarding the association between PA and CVF, many previous studies have dichotomized the participants into active or inactive based on the PA recommendation. In our study we found that meeting the PA recommendation was positively associated with CVF. These findings confirm the findings by Ortega et al. in 14–16-year-olds [39] and Marques et al. in 10–18-year-olds [21]. Considering that only a small proportion of adolescents meet the PA recommendations (30% in this sample), it is important to examine other aspects of PA. When using the continuous MVPA variable we found that MVPA was significantly positively associated with CVF levels and these findings have been established in previous studies with a greater age span [23,24]. However, there is a lack of studies including less intense PA, such as LIPA, as a predictor. We found that LIPA also was associated with higher CVF, however, it is important to note that LIPA was highly correlated to MVPA.

CVF can be enhanced by modifiable factors such as regular PA. However, the accelerometer data only give a snapshot of the current PA levels and although the reliability of several days of accelerometer recordings is reasonable, the data only capture a limited time span. Therefore, including other measures of PA habits gives additional data of PA patterns, of relevance to CVF levels in adolescents. One such measure used in this study was participation in organised sports where we found that adolescents who had participated in organised sports also had significantly higher CVF. Similar findings were seen in a longitudinal study, concluding that participants involved in sports had significantly higher CVF, spent more time in MVPA and less time in SED. However, participation rate was not investigated [40]. Our study showed that those who participated more than twice per week had significantly higher CVF compared to those who did not participate in organised sports. This indicates a stronger association to CVF in those who participate more frequently in organised sports. It is plausible to think that individuals who participate to a greater extent could have a longer history of participating in sports, either in multiple sports or at a higher level, which over time yields higher CVF levels. To our knowledge, no other study has investigated this association and future studies should further investigate the association between organised sports and CVF.

The finding that CVF was significantly higher in adolescents reporting participation in organised sports even after adjustment for MVPA is novel. However, it is important to note that other factors than PA, such as genetic influence, also contribute to the difference in CVF between participants and nonparticipants. This, in combination with the cross-sectional design, underlines the limitation of this study to determine a causal relation between participation in organised sports and CVF.

In our statistical models, we found that SED was negatively associated to CVF. Some previous studies have not found independent associations between SED and CVF when controlling for MVPA [23,24,28]. However, when using accelerometers to measure MVPA and SED, it could be problematic to include both outcomes in the same model due to multicollinearity. We therefore calculated a ratio between SED and MVPA time to account for both in the same model. Similar approaches have been used in other studies, which found that a combination of the two were associated with higher CVF but that MVPA was more important [24]. In our model we found our ratio to be significantly associated to lower CVF, which suggests that SED is also an important contributor to CVF.

Regarding screen time, we found that screen time on weekdays was significantly associated with CVF whereas weekend screen time was not a significant correlate. We found that there was significant difference in screen time between weekdays and weekend days. Porter el al. found no significant association between screen time and CVF [28]. One potential explanation for this could be that they did not differentiate between weekdays and weekends. It could be that high screen time on weekdays replaced other afterschool activities, such as organised sports. When we compared the groups, we found that those with higher screen time on weekdays also had significantly higher SED and boys also had significantly less time in MVPA. When controlled for SED, the association between weekdays screen time and CVF remained significant for girls and in boys with 5 h or more of screen time.

The strengths of the current study include a large sample size of Swedish adolescents, using detailed measures of PA (accelerometry), which provided a more detailed measure than self-reported PA. A limitation was using a submaximal VO_2_ fitness test to estimate CVF, which was less accurate than a VO_2_max test. However, the Ekblom-Bak test is better adapted for adolescents who are not used to being physically active, which potentially created a more representative study sample. Another limitation was the lack of questions covering non-organised PA, such as outdoor activities or free play. Although some studies suggest that participation in organised sports during adolescence is associated with higher PA during adulthood [18,19], organised sports are most prevalent during school-aged years. Therefore, covering other types of regular PA could have shed light on the potential association between non-organised PA such as outdoor activities and CVF since these types of activities potentially persist during adulthood. Due to the cross-sectional nature of this study causality cannot be determined. Therefore, future longitudinal studies are needed to examine the impact of different PA patterns on CVF over time.

## 5. Conclusions

This cross-sectional study showed that adolescents who participated in organised sports had significantly higher CVF and the association was stronger for those who participated more than twice per week. MVPA was also positively associated to CVF, whereas a high proportion of SED was negatively associated with CVF. There was a negative association in those who spent more than five hours on screen time on weekdays and CVF, in girls this was seen also in those with 3–4 h of screen time. Due to the cross-sectional nature of the study, causal relations remain unknown, however, the findings of this study could be relevant when designing future interventions with the aim to improve CVF.

## Figures and Tables

**Figure 1 children-08-00998-f001:**
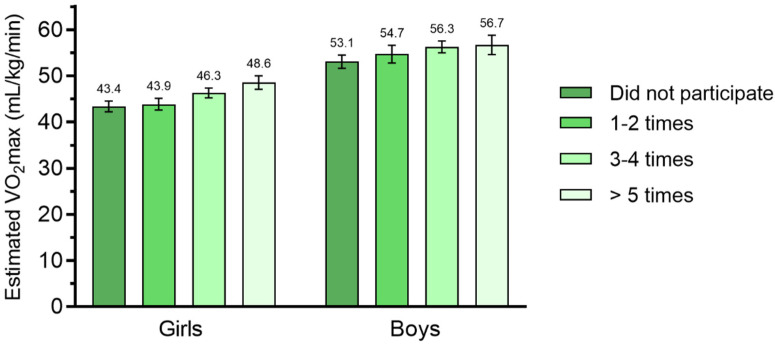
Adjusted mean CVF between adolescents participating in organised sports grouped into weekly participation rate. Values were adjusted for BMI, MVPA and wear time. Error bars represent 95% confidence interval.

**Table 1 children-08-00998-t001:** Descriptive characteristics of the study sample by sex (mean ± SD unless otherwise specified).

	All	Girls	Boys	Sig.
	n (%)	n (%)	n (%)	*p*
Demographics	1139 (100)	580 (51.0)	558 (49.0)	
Age (year)	13.4 ± 0.3	13.4 ± 0.3	13.4 ± 0.4	0.147
Parental education, ≥12 years n (%)	695 (71.6)	353 (71.0)	341 (72.2)	0.674
Participant country of birth, Sweden n (%)	967 (85.7)	490 (84.9)	476 (86.4)	0.758
Height	n = 1137	n = 580	n = 557	
Height (cm)	161.7	161.1	162.3	**0.007**
Weight	n = 1134	n = 580	n = 554	
Weight (kg)	52.8	53.1	52.6	0.491
BMI	n = 1134	n = 580	n = 554	
BMI	20.10 ± 3.6	20.40 ± 3.5	19.80 ± 3.6	**0.005**
BMI sds	0.36 ± 1.23	0.45 ± 1.11	0.26 ± 1.35	**0.012**
BMI Categories				0.203
Underweight n (%)	89 (7.8)	38 (6.6)	51 (9.2)	
Normal weight n (%)	815 (71.8)	430 (74.1)	384 (69.3)	
Overweight n (%)	179 (15.8)	89 (15.3)	90 (16.2)	
Obese n (%)	52 (4.6)	23 (4.0)	29 (5.2)	
Body composition	n = 1131	n = 580	n = 551	**<0.001**
Body fat (%)	22.6	25.7	19.3	
Physical activity	n = 903	n = 490	n = 413	
SED (min/day)	602.0 ± 66.6	608.9 ± 62.65	593.7 ± 70.2	**<0.001**
MVPA (min/day)	52.0 ± 19.0	49.5 ± 17.7	54.9 ± 20.1	**<0.001**
Light (min/day)	138.6 ± 30.2	137.5 ± 27.1	139.9 ± 33.6	0.231
Accelerometer wear time (min/day)	792.6 ± 60.8	796.0 ± 58.1	788.5 ± 63.8	0.067
Total valid days included for accelerometer	6.0 ± 1.1	6.2 ± 1.0	5.8 ± 1.1	**<0.001**
Reached the PA recommendations N (%)	273 (30.2)	121 (24.7)	152 (36.8)	**<0.001**
Organised sports				
Participated in organised sports N (%)	787 (72.0)	396 (71.4)	391 (72.7)	0.626
Participation rate (per week)				0.064
1–2 times N (%)	217 (28.0)	121 (30.9)	96 (25.0)	
3–4 times N (%)	356 (45.9)	181 (46.2)	175 (45.6)	
≥5 times or more N (%)	203 (26.2)	90(23.0)	113 (29.4)	
Screen time weekdays	n = 1125	n = 575	n = 550	0.842
≤2 h N (%)	359 (31.9)	179 (31.1)	180 (32.7)	
3–4 h N (%)	515 (46.0)	267 (46.4)	248 (45.1)	
≥5 h N (%)	251 (22.3)	129 (22.4)	122 (22.2)	
Screen time weekend	n = 1122	n = 576	n = 546	**<0.001**
≤2 h N (%)	178 (15.9)	70 (12.2)	108 (19.8)	
3–4 h N (%)	411 (36.6)	238 (41.3)	173 (31.7)	
≥5 h N (%)	533 (47.5)	268 (46.5)	265 (48.5)	
Fitness	n = 1020	n = 569	n = 451	
Estimated Vo_2_max (mL/kg/min)	49.5 ± 10.1	44.8 ± 8.5	55.5 ± 8.9	**<0.001**
Estimated Vo_2_max (L/min)	2.6 ± 0.02	2.3 ± 0.01	2.9 ± 0.03	**<0.001**
Puberty	n = 1041	n = 537	n = 504	0.346
Tanner score	3.26 ± 0.85	3.28 ± 0.79	3.23 ± 0.91	

BMI status according to IOTF 2012, BMI sds Karlberg et al., 2001. MVPA moderate-to-vigorous physical activity, PA physical activity. *p* are in bold to indicate significance (*p* < 0.05).

**Table 2 children-08-00998-t002:** Unadjusted mean fitness levels in different adolescent groups, stratified by gender.

	All			Girls			Boys		
		Vo_2_max (mL)	Sig.		Vo_2_max (mL)	Sig.		Vo_2_max (mL)	Sig.
	n	Mean ± SD	*p*	n	Mean ± SD	*p*	n	Mean ± SD	*p*
BMI status			**<0.001**			**<0.001**			**<0.001**
Underweight/normal weight	800	51.5 ± 9.4		460	46.8 ± 7.7		340	57.8 ± 7.6	
Overweight/obese	220	42.5 ± 9.8		109	36.4 ± 6.2		111	48.5 ± 8.9	
Parental education			**0.012**			0.168			**0.009**
≤12 years	249	48.6 ± 9.6		141	44.6 ± 7.7		108	53.8 ± 9.4	
>12 years	624	50.4 ± 9.9		348	45.8 ± 8.4		276	56.3 ± 8.4	
Country of Birth			**0.031**			**0.010**			0.868
Sweden	872	49.8 ± 10.0		481	45.2 ± 8.4		391	55.5 ± 8.9	
Outside Sweden	141	47.8 ± 10.5		85	42.7 ± 8.3		56	55.7 ± 8.4	
Puberty quartiles						**0.001**			**0.037**
1				258	46.1 ± 8.6		144	53.8 ± 8.3	
2				115	43.5 ± 8.4		75	56.5 ± 9.4	
3				93	44.2 ± 8.1		139	56.60 ± 9.03	
4				62	42.0 ± 7.3		63	55.5 ± 8.5	
Participated in organised sports			**<0.001**			**<0.001**			**<0.001**
Yes	700	51.0 ± 0.59		391	46.3 ± 0.4		309	57.1 ± 0.5	
No	280	46.0 ± 0.37		153	41.4 ± 0.6		127	51.6 ± 0.8	
Reached the PA recommendations			**<0.001**			**<0.001**			**0.001**
Yes	305	53.1 ± 0.6		144	48.0 ± 0.4		161	57.7 ± 0.6	
No	682	47.9 ± 0.4		410	43.8 ± 0.7		272	54.1 ± 0.5	

BMI status according to IOTF 2012. PA physical activity. *p* are in bold to indicate significance (*p* < 0.05).

**Table 3 children-08-00998-t003:** Associations between predictors and fitness analysed with linear regression models controlled for BMI, and wear time, stratified by gender.

Model	Girls			Boys		
	n	β	95% CI	n	β	95% CI
1. Physical activity	482			346		
1.1 MVPA whole week		0.074	**0.038, 0.110**		0.069	**0.029, 0.108**
1.2 SED whole week		−0.039	**−0.057, 0.022**		−0.030	**−0.046, −0.014**
1.3 SED:MVPA ratio		−0.473	**−0.699, −0.248**		−0.407	**−0.638, −0.175**
1.4 Light PA whole week		0.043	**0.020, 0.066**		0.035	**0.013, 0.057**
1.5 Reached the PA recommendations		3.313	**1.962, 4.664**		1.874	**0.262, 3.486**
2. Organised sports						
2.1 Participated in organised sports	461	2.269	**0.853, 3.686**	341	2.870	**1.141, 4.599**
3. Rate of participation (per week)						
Do not participate at all		REF			REF	
1–2 times		0.486	−1.197, 2.170		1.635	−0.763, 4.032
3–4 times		2.937	**1.302, 4.572**		3.120	**1.242, 5.154**
≥5 times or more		5.188	**3.262, 7.115**		3.615	**1.099, 6.130**
4. Screen time weekdays	458			428		
≤2 h		REF			REF	
3–4 h		−1.824	**−3.209, −0.440**		−1.579	−3.356, 0.198
≥5 h		−2.315	**−3.922, −0.708**		−2.821	**−5.140, −0.502**
5. Screen time weekend	479			341		
≤2 h		REF			REF	
3–4 h		0.582	−1.543, 2.708		1.308	−0.836, 3.452
≥5 h		−1.266	−3.362, 0.831		−1.138	−3.238, 0.963
6. Body composition	569			448		
Body fat (%)		−0.520	**−0.671, −0.369**		−0.714	**−0.887, −0.541**
7. Puberty	528			421		
Tanner		−0.621	−1.349, 0.108		2.613	**1.860, 3.366**

CI are in bold to indicate significance (*p* < 0.05). Model 2, 3, 4 and 5 are also adjusted for MVPA. MVPA moderate-to-vigorous physical activity, PA physical activity, SED sedentary time.

## Data Availability

The datasets are not available for download in order to protect the confidentiality of the participants. The data are held at The Swedish School of Sport and Health Sciences.

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
