# Peer review of "Associations between Physical Activity Patterns, Screen Time and Cardiovascular Fitness Levels in Swedish Adolescents"

_children, 2021, doi:10.3390/children8110998_

Round 1

Reviewer 1 Report

This is a high quality article and the introduction, methods, and results are presented in a clear manner. The findings are sound and are presented nicely. My only critique is that opportunities for organized sport are most prevalent during school-aged years, but what was not addressed in this article was background literature supporting whether  participation in organized sport during youth will translate to lifelong physical activity during adulthood, or if the CVF health benefits are temporary. I include this critique because sport is usually focused on competition whereas outdoor play, and lifelong physical activities may be more sustainable. I did not see anything related to outdoor activities or free play included in the article. 

Reviewer 2 Report

The manuscript entitled "Associations between physical activity patterns, screen time and cardiovascular fitness levels in Swedish adolescents" is a great manuscript. The brief introduction is well structured, and it contains all the necessary information to understand the study. The "Material and Methods" had detailed information about the study. The "result" section only concentrating on the main result, which has appropriate statistical methods. The authors used several empirical supports in their discussion, and their conclusion is adequate.

Overall, it is an excellent study.
